# Conformational Changes in Ff Phage Protein gVp upon Complexation with Its Viral Single-Stranded DNA Revealed Using Magic-Angle Spinning Solid-State NMR

**DOI:** 10.3390/v14061264

**Published:** 2022-06-10

**Authors:** Smadar Kedem, Roni Rene Hassid, Yoav Shamir, Amir Goldbourt

**Affiliations:** School of Chemistry, Tel Aviv University, Ramat Aviv, Tel Aviv 6997801, Israel; smadark@cv.technion.ac.il (S.K.); ronihassid@tauex.tau.ac.il (R.R.H.); yoavshamir5@gmail.com (Y.S.)

**Keywords:** solid-state NMR, fd bacteriophage, DNA binding protein, gVp, protein–DNA interactions

## Abstract

Gene V protein (gVp) of the bacteriophages of the Ff family is a non-specific single-stranded DNA (ssDNA) binding protein. gVp binds to viral DNA during phage replication inside host *Escherichia coli* cells, thereby blocking further replication and signaling the assembly of new phage particles. gVp is a dimer in solution and in crystal form. A structural model of the complex between gVp and ssDNA was obtained via docking the free gVp to structures of short ssDNA segments and via the detection of residues involved in DNA binding in solution. Using solid-state NMR, we characterized structural features of the gVp in complex with full-length viral ssDNA. We show that gVp binds ssDNA with an average distance of 5.5 Å between the amino acid residues of the protein and the phosphate backbone of the DNA. Torsion angle predictions and chemical shift perturbations indicate that there were considerable structural changes throughout the protein upon complexation with ssDNA, with the most significant variations occurring at the ssDNA binding loop and the C-terminus. Our data suggests that the structure of gVp in complex with ssDNA differs significantly from the structure of gVp in the free form, presumably to allow for cooperative binding of dimers to form the filamentous phage particle.

## 1. Introduction

Bacteriophages are the most abundant organism found in the biosphere. Since their discovery almost simultaneously a century ago by Fedrick Twort and Felix d’Herelle, bacteriophages have been the focus of a considerable body of scientific research. They are considered alternative antibacterial agents to battle infections [1], are relevant for food safety [2] and preserving meat and vegetables, and are also key in agriculture [3] and aquaculture [4]. Although most bacteriophages kill their bacterial host during their replication process, the filamentous bacteriophages of the Ff family (M13, fd, and f1) replicate episomally.

Ff phages have over 98% sequence identity, with identical gene V protein (gVp, also termed pV) sequences and a single site mutation (D12N) on the surface of the major coat protein of M13. The mature phages have a rod-like structure of 6–7 nm in diameter and approximately 1 μm in length consisting of thousands of identical subunits of pVIII, the major coat protein, which forms a helical structure with a five-fold symmetry axis and an approximate twofold screw axis C_5_S_~2_ (class I) [5,6]. These proteins encapsulate a circular single-strand of DNA consisting of several thousands of nucleotides and a few copies of minor proteins cap the ends of the particle [7].

During the life cycle of the filamentous phage, the viral ssDNA is injected into the host cytoplasm via the F-pilus lumen [8]. Phage proteins are stored in the inner cell membrane of the bacteria and the viral genome is transformed into a double-stranded replicative form DNA (RF DNA) by host enzymes. mRNA transcription is initiated from constitutive promoters on the RF DNA, leading to the expression of all eleven phage-encoded proteins. Simultaneously, single-stranded copies of the viral DNA are generated via the rolling circle replication mechanism using the RF as a template. As the number of RF DNA molecules increases, so does the concentration of the phage proteins present in the host cell. When gVp reaches a threshold concentration in the host cell, it binds cooperatively to a hairpin on the ssDNA and coats its entire length with approximately 1500 proteins per ssDNA, thereby forming an intermediate complex that prevents conversion into RF DNA [9]. gVp is essential for progeny ssDNA production [10]. The gVp molecules in this intermediate form are then exchanged for the major coat protein subunit and the four minor coat protein subunits, resulting in a mature viral phage particle [11,12]. The gVp molecules are recycled into the cell plasma and are able to coat another strand of viral ssDNA [11,12].

The gVp encoded by Ff phage has 87 amino acids and it forms a stable homodimer over a wide range of ionic strengths, pHs, and temperatures [13]. The structure of the dimer was solved using both X-ray crystallography [14] and solution NMR spectroscopy [15,16] (Figure 1). The protein structure involves a distorted six-stranded β-barrel (see topology in [17]) and has three main loops. The loop that connects the second and third beta strands is associated with DNA binding and is termed the ‘DNA binding loop’. The loop that connects the third and fourth beta strands participates in dimer contacts and partially in the creation of a dimer–dimer interface and is termed the ‘complex loop’. The third, so-called ‘dyad loop’, connects beta strands 6 and 7; this loop is not part of the core beta barrel but protrudes toward the symmetry-related monomer and participates in the dimer interface, as well as in cooperative protein–protein binding interactions. Although gVp is classified as a non-sequence-specific single-stranded DNA-binding protein, it can also bind to double-stranded DNA, double-stranded RNA, and G-quadruplex structures [18,19,20,21]. Studies showed that gVp binds to ssDNA strands that are conjugated to gold nano-particles and that gVp mediates the interaction of non-complementary ssDNA strands; the interactions are abolished if a complementary ssDNA is introduced [22,23].

Transmission electron microscopy measurements and scattering techniques revealed that approximately 1500 molecules of gVp bind as dimers to two anti-parallel ssDNA strands to form a left-handed helical complex with a pitch in the range of 6–12 nm and lengths in the range of 0.76–0.89 μm [25,26,27]. Several hundred of these complexes can be present per cell [11]. Circular dichroism and fluorescence spectroscopy measurements indicated that the protein monomer binds to between three to five nucleotides, depending on the solution ionic strength [28,29,30,31]. Low-angle neutron scattering experiments indicate that the ssDNA is located near the center of the complex and has a cross-sectional radius of gyration of about 18 Å, whereas the protein has a cross-sectional radius of gyration of about 33 Å [32].

The binding of gVp to oligonucleotides was extensively studied using solution NMR. Data suggest that electrostatic interactions, which involve ion pairing of positively charged lysine and arginine protein side-chains with the negatively charged ssDNA phosphate groups, stabilize the complex [33]. Additionally, NMR experiments indicate that hydrophobic stacking interactions occur between leucine, tyrosine, and phenylalanine residues of the protein and the DNA bases [34,35,36].

Despite these data and efforts to model the entire nucleoprotein complex [12,17], an accurate experimentally supported atomic-resolution structure of the complex has yet to be established. It also remains unknown whether and how the protein undergoes structural changes in order to facilitate ssDNA binding. A recent study showed that gVp does undergo structural changes when exposed to ectoine, which is an osmolyte that stabilizes proteins in extremophilic microorganisms; however, these changes resulted in reduced binding to ssDNA [37].

Magic-angle spinning solid-state NMR spectroscopy provides detailed atomic-resolution information on proteins, polynucleic acids, and protein complexes [38,39,40]. Magic-angle spinning solid-state NMR is particularly advantageous when studying systems of high molecular weight, or of low solubility, or systems that do not result in high-quality crystals or that cannot be clearly resolved in cryo-electron microscopy. Solid-state NMR is also useful for a system that exhibits dynamics or intrinsic disorder that prevents characterization using diffraction techniques. These properties make solid-state NMR an excellent technique for studying intact viruses, viral capsids, and nucleocapsids [41,42,43,44].

In this study, we utilized solid-state NMR to provide insight into the structural changes that gVp undergoes upon the formation of a complex with full-length ssDNA originating from the fd phage. We performed a de-novo chemical shift assignment of the protein in the complex and compared these shifts to those of the free form of gVp. We also compared the chemical-shift-derived secondary structure to that of the X-ray structure of a crystalline free gVp. Our data showed that there are considerable structural changes in gVp upon complexation with ssDNA, particularly at the ssDNA binding loop and in the C-terminal region.

## 2. Materials and Methods

### 2.1. Sample Preparation

The complex was prepared in four steps; (i) overexpression and purification of gVp, (ii) production of fd viruses, (iii) extraction of ssDNA, and (iv) complexation.

(i) The expression and purification of gVp were described in an earlier work [45]. In brief, we used a pET-30b plasmid with kanamycin (KAN) resistance to overexpress gVp with an N-terminal His-tag. The plasmid was transformed into *Escherichia coli* BL21(DE3)-competent cells. In order to obtain labeled proteins, we used the Bracken protocol for expression in minimal media [46]. A starter culture was grown in 1 L LB media (37 °C, 220 rpm, 50 μg/mL KAN) until the log phase. For the preparation of natural abundance samples, the culture was immediately induced with 1 mM isopropyl β-D-1-thiogalactopyranoside (IPTG). Samples to be labeled were centrifuged. The cell pellet was washed with 150 mL M9 stock solution; centrifuged; and resuspended in 250 mL buffer solution containing M9, metal trace, 2 mM MgSO_4_, 50 μg/mL KAN, BME vitamin stock (Sigma-Aldrich, Saint Louis, MO, USA), 4 g/L fully ^13^C-labeled D-glucose, and 1 g/L ^15^NH_4_Cl. Sparsely labeled samples were prepared by replacing the ^13^C-D-glucose with 1,3-^13^C glycerol (2 g/L) and NaHCO_3_ at natural abundance (2 g/L). After shaking for an hour at 220 rpm, we added 1 mM IPTG in order to induce protein expression, and cell growth was continued for an additional 20 h at 27 °C (220 rpm). For purification of both labeled and unlabeled samples, cells were centrifuged and then dissolved in a 25 mL buffer A (200 mM NaCl, 20 mM imidazole, 50 mM Tris-HCl pH 7.4) with DNase (Merck, Burlington, MA, USA) and 10 mM MgCl_2_. The cells were lysed using a tissue homogenizer and then using a Microfluidics M-110L microfluidizer. After pelleting the cell debris, the supernatant was loaded onto a metal ion affinity chromatography (10 mL HisTrap HP prepacked with Ni Sepharose). The column was washed with buffer A supplemented with 1 mM β-mercaptoethanol and then with buffer B (1.0 M NaCl, 50 mM Tris-HCl pH 7.4, 1 mM β-mercaptoethanol) to remove DNA fragments, and the protein was eluted using buffer C (200 mM NaCl, 300 mM imidazole, 50 mM Tris-HCl pH 7.4, 1 mM β-mercaptoethanol). The purified protein was dialyzed using SnakeSkin dialysis tubing (Thermo Scientific, San Jose, CA, USA) against a complexation buffer containing 200 mM NaCl, 1 mM EDTA, and 10 mM Tris-HCl pH 7.4.

A gel demonstrating protein purity is shown in Appendix A. Purified gVp was stored after dialysis in a 10 mM tris-buffer containing 200 mM NaCl and 1 mM EDTA at a pH of 7.4 (complexation buffer). UV characterization revealed a maximum signal at 276 ± 1 nm, in agreement with prior observations [47]. The protein has a molecular weight of 11,079.76 Da and contains 87 amino acids, as well as an NMR-invisible 12-residue-long C-terminal His-tag tail (in parentheses):

MIKVEIKPSQ AQFTTRSGVS RQGKPYSLNE QLCYVDLGNE YPVLVKITLD

EGQPAYAPGL YTVHLSSFKV GQFGSLMIDR LRLVPAK(LAAALEHHHHHH)

(ii) The preparation and purification of fd bacteriophages containing the extended genome fth1 (8233 nucleotides, Mw 2.542 MDa) with tetracycline resistance [48] were previously reported [49].

(iii) The circular ssDNA was isolated from fd using the phenol-chloroform extraction technique. We added 10 mL of phenol:chloroform:isoamyl alcohol (25:24:1), pH 7.5–8.0 solution (Sigma-Aldrich) to an equal volume of fd virus solution at a concentration of ~1 mg/mL. The suspension was vigorously vortexed for 1 min and then centrifuged at 14,000 rpm for 20 min at 4 °C. The upper aqueous layer containing the target ssDNA was transferred to a clean tube, and the separation process was repeated until no layer of protein was observed between the organic and the aqueous phases. Then, 0.1 M sodium acetate solution (pH 5.7) and 2.5 volumes of ice-cold absolute ethanol (final *v*/*v* concentration of 72.5%) were added to the aqueous layer. The mixture was incubated at −20 °C overnight (or for 2 h at −80 °C) to precipitate ssDNA, centrifuged at 14,000 rpm for 20 min at 4 °C, and the supernatant was removed from the pellet. The pellet was washed with 500 μL of 70% ethanol, followed by centrifugation at 14,000 rpm for 20 min at 4 °C, dried overnight, and dissolved in the complexation buffer. The purity of the ssDNA was verified by running the extracted ssDNA on 1.0% agarose gel electrophoresis. The total yield of the ssDNA extracted from ~10.0 mg fd phage was ~1.0 mg.

(iv) We prepared the gVp ssDNA complex by mixing gVp and ssDNA in a mass ratio of 9 to 1 at room temperature for 20 min. Under these conditions, the binding stoichiometry is reported to be four nucleotides per gVp monomer [27]. Complexation was assessed using UV measurements, agarose gel electrophoresis, and solid-state NMR rotational echo double resonance (REDOR) measurements.

In preparation for magic-angle spinning solid-state NMR measurements, samples of the complex were precipitated with 10% PEG and 5 mM MgCl_2_. Free gVp samples were precipitated using 30% PEG and 10 mM MgCl_2_. Following several centrifugation steps into sealed 200 μL tips, the precipitated samples were transferred to a 4 mm ZrO_2_ rotor.

### 2.2. Gel Electrophoresis Binding Assay

Different ratios of ssDNA to gVp ranging from 3 to 50 nucleotides per monomer were analyzed on an agarose gel run in 168 mM Tris-base, 80 mM NaOAc, 7.2 mM disodium EDTA dihydrate, pH 8.3 (GBB buffer).

### 2.3. NMR Experiments

Magic-angle spinning solid-state NMR experiments were performed using a Bruker Avance III spectrometer connected to a wide-bore superconducting magnet with a magnetic field of 14.1 T. An Efree probe operating in HCN triple-resonance mode was used for most multi-dimensional NMR experiments. Protein-ssDNA distances were obtained using the ^31^P{^13^C} REDOR experiment [50]. A 4 mm DVT probe operating in HPC mode was used, and the temperature was controlled at the entrance to the probe. Data were processed using nmrPipe and analyzed with the SPARKY program. Additional experimental details are given in the figure captions and in the Appendix A. The REDOR curve was fit to simulations obtained with the SIMPSON software [51].

## 3. Results and Discussion

### 3.1. Complex Formation between gVp and fd Phage ssDNA

We initially characterized the formation of the complex formed by purified fd gVp and the circular, 8233-nucleotide-long ssDNA isolated from fd phage using agarose gel electrophoresis (Figure 2). The free ssDNA migrated on this type of gel according to its molecular weight. At a relatively low binding ratio of 50 nucleotides per monomer of gVp, migration of the ssDNA was slightly retarded relative to that of ssDNA in the absence of protein resulting from an increase in molecular weight and smaller available charge. As the ratio of nucleotides to gVp monomer was decreased and more proteins bind the ssDNA, the ssDNA migrated more slowly. Little change in mobility was observed when the ratio decreased below 5 nucleotides per monomer, indicating that the binding saturated at 3–5 nucleotides per monomer.

We also analyzed solutions of purified gVp and ssDNA mixed at different ratios using UV measurements. There was a shift from a maximum at 276 nm for the free gVp to a maximum at 258 nm for the nucleoprotein complex (Appendix A), in accordance with the observations of Day et al. [47].

The complexation was also assessed directly with solid-state NMR in two ways. First, we conducted two two-dimensional (2D) dipolar assisted rotational resonance (DARR) experiments on samples of (i) uniformly labeled gVp complexed with ssDNA with isotopes at natural abundance (U-gVp/NA-DNA) and (ii) unlabeled gVp complexed with uniformly labeled ssDNA (NA-gVp/U-DNA). Figure 3 shows the spectrum of U-gVp/NA-DNA in overlay with that of NA-gVp/U-DNA. The spectrum of the complex with labeled gVp shows predominantly protein signals, whereas that with labeled ssDNA is characterized by ribose and base cross-peaks typical of DNA and a clear signature of the thymine methyl groups and their contact with the ribose. Inspection of the diagonal of the spectrum NA-gVp/U-DNA revealed many protein signals that arise from the natural abundance of gVp. Those signals, predominantly located at the aliphatic and carbonyl regions, lack cross-peaks in the spectrum of NA-gVp/U-DNA but are present in the spectrum of the complex formed with labeled proteins.

The spectra in Figure 3 show the existence of protein and DNA signals (similarly to the UV titration spectra). To directly evaluate the correlation between the DNA and gVp proteins, we conducted ^31^P-^13^C REDOR experiments. In REDOR, the difference in ^31^P signal intensity, which was obtained with (S) and without (S_0_) ^13^C π pulses, depends to an excellent approximation on the ^31^P-^13^C dipolar couplings. Therefore, by collecting a set of experiments in which the number of pulses is increased and subsequently fitting the REDOR fraction (S_0_-S)/S_0_ to simulations, the ^31^P-^13^C distance can be elucidated. Figure 4 shows the REDOR fraction curve and the best fit SIMPSON simulation using three carbons at a distance of 5.5 Å from the phosphorous. Including more than three carbon spins in the simulation did not alter our result, and fewer carbons resulted in a worse fit. It is likely that the curve arises from a combination of 1-3 carbons since the protein is uniformly labeled, and therefore, 5.5 Å is an average typical distance from, for example, the three carbons at the ring of a tyrosine amino acid to the phosphorous backbone of the ssDNA.

### 3.2. Chemical Shift Assignment of gVp

To assess the structural similarities between free and bound gVp, we assigned the chemical shifts of the gVp complex by conducting two- and three-dimensional experiments on U-gVp/NA-DNA. An additional sample was prepared with sparsely labeled gVp by supplementing the minimal media with 1,3-^13^C glycerol as the sole carbon source, followed by complexation with NA-DNA (gly13-gVp/NA-DNA). The assignment was based on dipolar-based 2D radio-frequency-driven recoupling (RFDR) [52], 2D combined $R2_{n}^{v}$-driven recoupling (CORD) [53], DARR [54], 2D NCO and NCA using double cross-polarization [55,56], and 3D NCACX and NCOCX [57]. Additional data was obtained using a scalar-coupling-based z-filtered refocused INADEQUATE experiment [58], which provided further validation of many of the assigned chemical shifts, and revealed some additional chemical shifts belonging to dynamic residues that were missing in dipolar-based experiments.

Figure 5 shows a typical 2D spectrum and projections from the 3D spectra of the U-gVp/NA-ssDNA sample recorded on a 600 MHz spectrometer with two examples of connectivities for cysteine-33 and serine-67. The overall resolution was sufficient for the assignment. We also conducted 2D and 3D experiments on the gly13-gVp/NA-DNA sample. Experiments on this sample had better spectral resolution and validated assignments of sidechains. An overlay of two DARR spectra acquired with a mixing time of 100 ms is shown in Appendix A; this overlay demonstrates the reduced spectral congestion in the gly13-gVp/NA-DNA sample compared with that in the U-gVp/NA-ssDNA sample.

Figure 6 shows strip plots connecting NCACX and NCOCX spectra for residues 9–15; similar plots for all other regions of the protein are shown in Appendix A. These plots demonstrate the quality of the sequential connectivities in gVp.

Although the dipolar-based experiments provided assignments for most of the protein residues, signals from some amino acids could not be detected or could only be detected at very long mixing times with weak signal-to-noise due to dynamics. To probe these regions, we employed the z-filtered refocused (zfr) INADEQUATE experiment. One-bond contacts from spin systems belonging to various amino acids, which were absent in the dipolar-based experiments, exhibited strong cross-peaks in the zfr-INADEQUATE spectrum (Figure 7).

Overall, we assigned 98% of N, Cα, Cβ, and CO chemical shifts and 90% of all possible ^13^C and ^15^N chemical shifts of gVp using the described experiments. The shifts were deposited in the BMRB under accession number 51391 and are also listed in Appendix A. These data allowed us to perform a thorough chemical shift-based comparison of the structure of gVp bound to ssDNA to that of the free gVp.

### 3.3. Comparison with Free gVp

The structure of free gVp was solved using both X-ray crystallography and solution NMR. In its crystal form, gVp is a homodimer with the two units related by a two-fold symmetry axis. It contains eight beta-strands, six of which are involved in a distorted antiparallel beta-barrel defining the monomer’s hydrophobic core. Three main loop regions protrude from this core: the dyad loop (residues 68–78), the DNA binding loop (residues 13–31), and the complex loop (residues 36–43). Another loop (termed the ‘broad connecting loop’ [17]) involves residues 48–59. It starts after the fourth beta strand at the end of the complex loop and positions the fifth strand at the edge of the internal β-sheet barrel.

We previously assigned the chemical shifts of the free form of gVp [45]. A comparison with the spectrum of gVp in the complex revealed distinctive chemical shift perturbations (CSPs) due to the presence of the ssDNA (Figure 8). Some of these changes were very large. For example, the CSP of T15 is 3.8 ppm. The sequential T14–T15 contacts validating the assignment in the free form are weak cross-peaks below T14 (T15Cβ–T14Cα) and above T15 (T14Cβ–T15Cα) in Figure 8. The validation of the same shifts of the bound form was somewhat more complex and is shown in the inset of Appendix A, where the contact of T15Cβ (at 68.9 ppm) with that of R16Cα (at 53.9 ppm) is a weak signal.

To further analyze the CSPs, we calculated for each amino acid the root-mean-square deviation (RMSD) of the weighted sum of the N, CO, Cα, and Cβ shifts of the bound and unbound forms of gVp. The results are presented in Figure 9, and independent CSPs are plotted for each backbone atom in Appendix A. The RMSDs are substantial over the entire protein (1.3 ± 1.0 ppm), suggesting a significant structural change upon binding. The most significant changes are in the DNA binding loop (residues 13–31), with RMSDs of 2.0 ± 1.1 ppm, and in the C-terminus (residues 81–86), with RMSDs of 2.4 ± 0.9 ppm. Interestingly, residues 36–40 in the complex loop show relatively small RMSDs of 0.9 ± 1.0 ppm (residues 41 and 42 were unassigned, and V43 is ambiguous with V45), suggesting that the interactions detected in the free dimer in this region are preserved in the form bound to DNA. Residues in the dyad loop (amino acids 68–78) have RMSDs of 1.2 ± 0.7, which is below the total average but above an average value of 0.9 ± 0.7, which does not include the DNA binding loop, the dyad loop, or the C-terminus. Another unique region with large RMSDs includes residues L49-D50-E51 at the beginning of the fourth loop. The large RMSDs of this region suggest that this loop undergoes structural adjustment to allow for DNA binding. In a previous model of the entire complex [17], residues D50 and Q53 (as well as H64, S66, and R82) were shown to participate in the formation of dimer–dimer contacts, interacting with E40 and Y41 (chemical shifts for those residues were undetected in our data) of another dimer. Another interaction region involved residues in the dyad loop (K69, Q72, F73, and D79).

A further structural comparison of free and bound gVp was performed using torsion-angle prediction methods that are based on the chemical shifts. We used TALOS-N [59] to predict the torsion angles of bound gVp and compared the predicted values to those extracted from the X-ray structure of free gVp. In order to locate regions with significant deviations, we plotted the quadratic average of the torsion angle differences, given by $\sqrt{0.5\left(\Delta \phi^{2}+\Delta \psi^{2}\right)}$, per residue (Figure 10). The five-point moving average revealed regions with significant deviations, which are those predicted to undergo structural changes upon DNA binding. This plot emphasizes that there are significant (*p* < 0.01) differences in the region of the DNA binding loop, in particular when comparing the mean value of 30 degrees obtained for residues 10–31 to the mean value of 10 degrees for residues 52–71; the latter residues have small RMSDs for their CSPs and belong to the broad connecting loop and β-strands five and six, ending just before the dyad loop. As was the case for RMSD calculations based on CSPs, we detected a large change in the orientation of residue 49, and only slightly smaller changes in residues 50 and 51 (these two residues had low reliability in TALOS), in agreement with the RMSD values extracted from the CSPs of those three residues.

## 4. Conclusions

The ssDNA binding protein gVp of fd bacteriophage binds cooperatively to the viral ssDNA. Previous studies revealed atomic-resolution details of the homodimer structure in the form not bound to DNA and a model of the complex identified potential ssDNA binding residues. Nonetheless, the atomic details of the gVp structure in the bound form remained elusive. In the current study, we used solid-state NMR to demonstrate that gVp undergoes extensive structural changes upon complexation with circular ssDNA isolated from the fd phage. The changes occurr throughout the protein but are most extensive in the ssDNA binding loop, in an exposed broad connecting loop positioning one of the strands of the beta barrel, and in the C-terminus. Our data suggested that the structure of gVp in the free form differs considerably from that of the protein bound to DNA. Presumably extensive structural alterations are required for binding to ssDNA and in order to allow cooperative binding of dimers to form the entire filamentous premature phage particle.

## Figures and Tables

**Figure 1 viruses-14-01264-f001:**
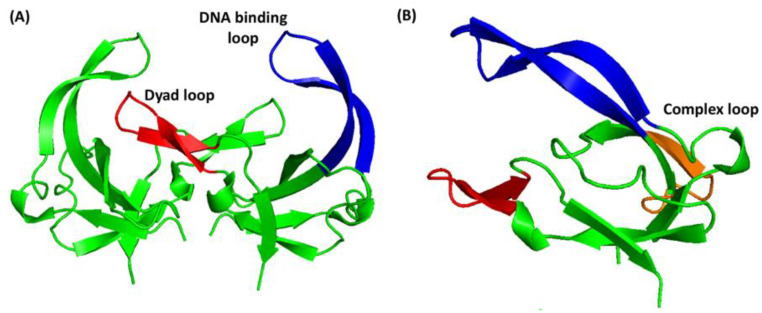
Ribbon diagram of gVp structure as determined using solution NMR [16]. (**A**) The gVp dimer with a symmetry axis parallel to the plane of illustration. (**B**) View of the gVp monomer rotated around the Y-axis on panel A. The three major loops, the dyad loop (red), the complex loop (orange), and the DNA binding loop (blue), are indicated. The images were reproduced from data stored in the Protein Data Bank, pdb id 2GVB, using the software PyMOL [24].

**Figure 2 viruses-14-01264-f002:**
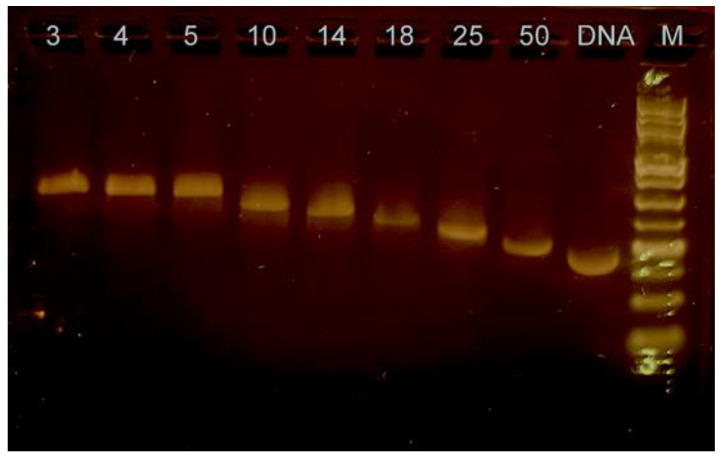
Agarose gel analysis of complexation of gVp with the circular ssDNA from the fd phage. The numbers above lanes indicate the ratio of nucleotides to protein monomers. The gel was stained with SYBR^TM^ GOLD (Invitrogen, Waltham, MA, USA). ‘DNA’ indicates the lane with fd ssDNA in the absence of protein. ‘M’ indicates the lane with the DNA ladder (GeneDireX, Taoyuan, Taiwan) with bands corresponding from bottom to top to 0.250, 0.5, 0.75, 1.0, 1.5, 2.0, 2.5, 3.0, 4.0, 5.0, 6.0, 8.0, and 10.0 kbp. The image was slightly processed to reduce brightness and increase visibility.

**Figure 3 viruses-14-01264-f003:**
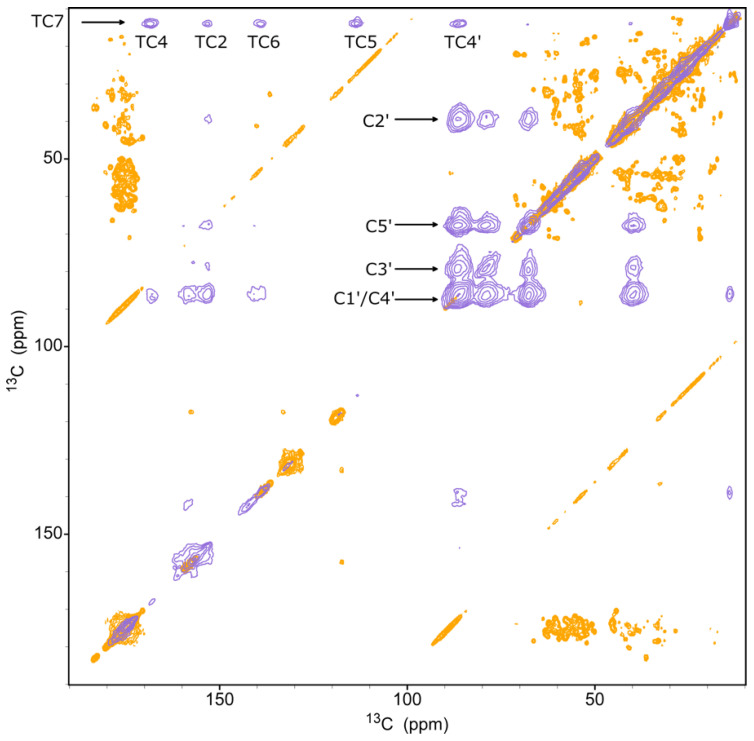
Spectral overlay of two 2D ^13^C-^13^C DARR spectra of U-gVp/NA-DNA (orange) and of NA-gVp/U-DNA (blue). The mixing times were 5 ms and 100 ms, respectively. Signals of the unlabeled protein are observed on the diagonal of the blue spectrum, where many correlations are observed in the orange spectrum, including the aliphatic and carbonyl protein-unique regions. Signals due to correlations between ssDNA ribose sugar carbons (C1′–C5′) and of the thymine methyl group (TC7) with other base carbons (TC2, TC4, TC5, TC6) and with C4′ are marked. Weaker thymine base–sugar correlations with C1′/C4′ are also visible.

**Figure 4 viruses-14-01264-f004:**
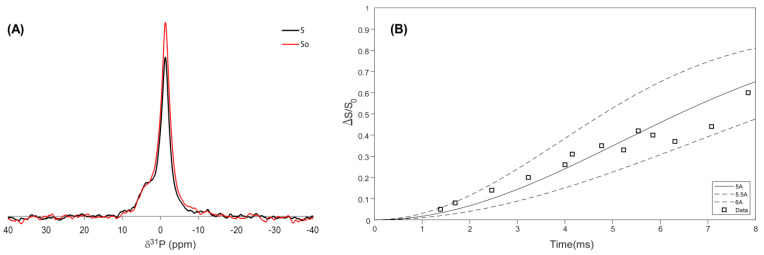
REDOR analysis of gVp-ssDNA complex. (**A**) ^31^P S (black) and S_0_ (red) REDOR signals of gVp-ssDNA complex that were obtained at a mixing time of 3.23 ms. (**B**) ^31^P{^13^C} REDOR fraction curve and SIMPSON simulations showing the best fit at 5.5 ± 0.5 Å. Simulations were obtained for a four-spin system consisting of one phosphorous spin and three carbon spins. ^31^P shifts are referenced relative to O-phospho-L-Serine at 0.3 ppm.

**Figure 5 viruses-14-01264-f005:**
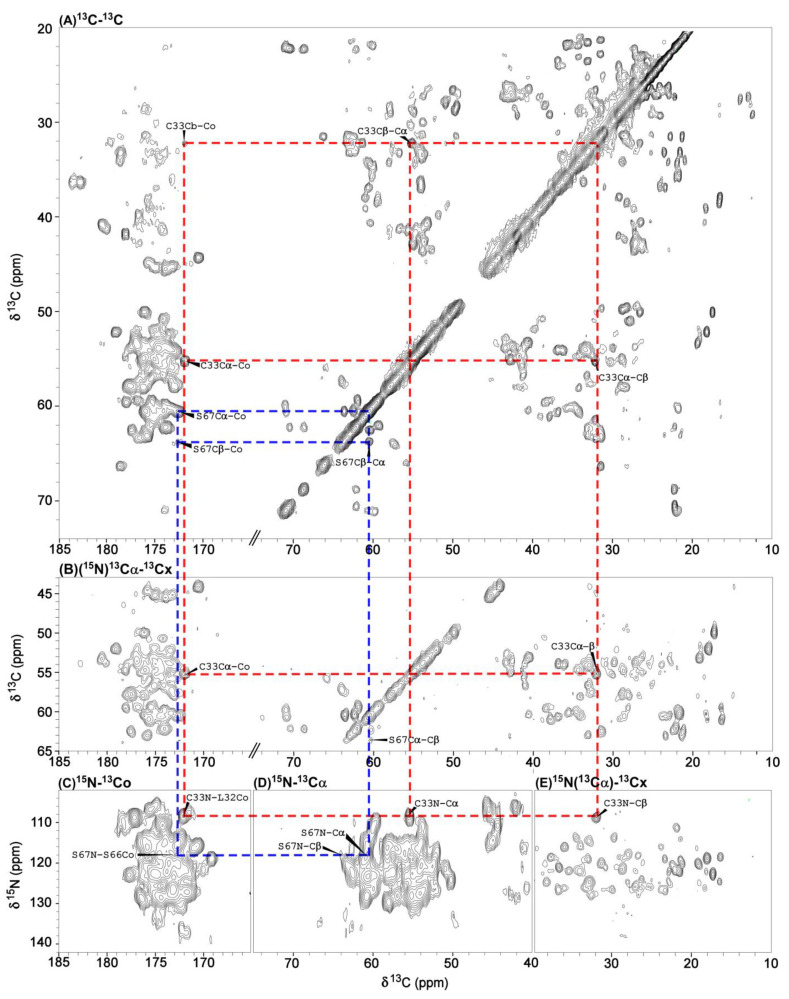
Four different spectra of U-gVp/NA-DNA complex. (**A**) The 2D ^13^C-^13^C DARR spectrum acquired with a 5 ms mixing time (DARR5); (**B**) 2D (^15^N)^13^Cα^13^Cx projection from 3D NCACX25; (**C**) 2D ^15^N-^13^CO; (**D**) 2D ^15^N-^13^Cα; (**E**) ^15^N(^13^Cα)^13^Cx projection from the 3D NCACX25 experiment. Sequential assignments of S67 and C33 are indicated using blue and red dashed lines, respectively. All spectra were processed with a sine-squared window function.

**Figure 6 viruses-14-01264-f006:**
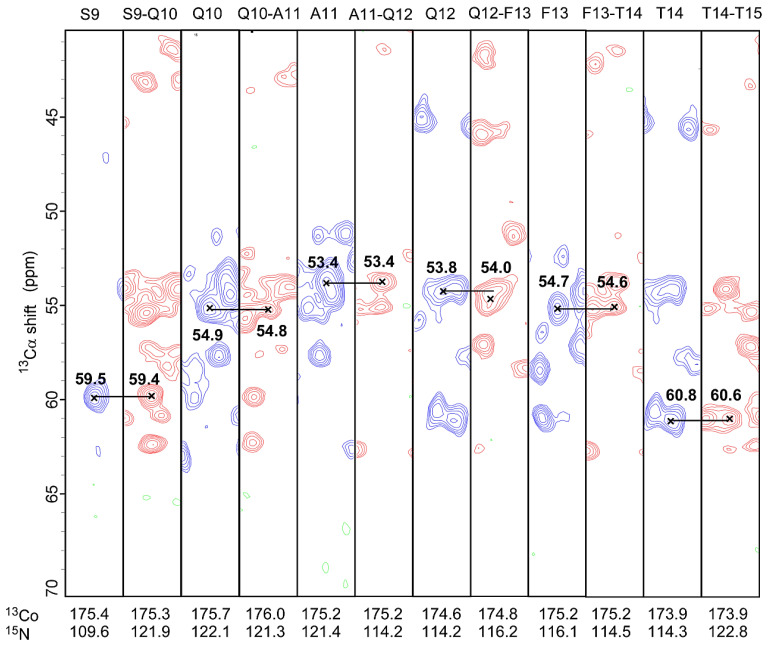
Representative strip plots of residues 9-15 from U-gVp/NA-DNA complex. Sequential assignments from the 3D spectra of NCACX (blue) and NCOCX (red) are illustrated. Correlations with matching Cα and CO chemical shifts are connected with solid lines.

**Figure 7 viruses-14-01264-f007:**
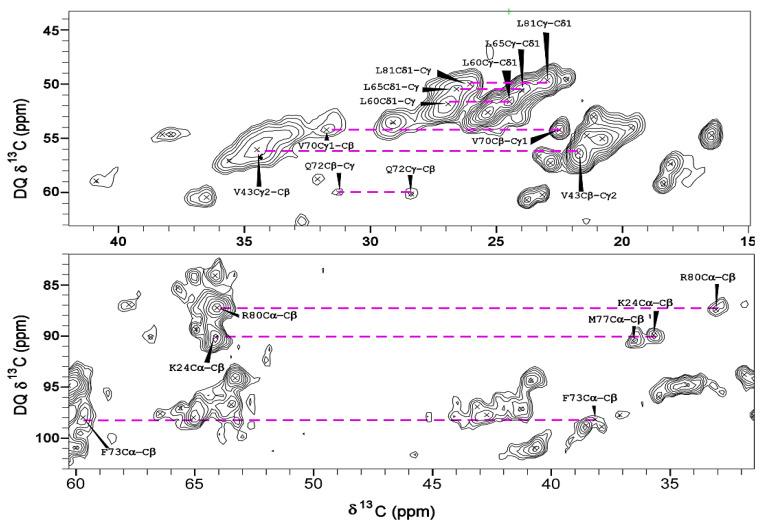
^13^C zfr-INADEQUATE 2D spectrum of U-gVp–NA-DNA complex. The spectrum depicts one-bond connectivities (dashed lines) at their sum frequencies in the double-quantum indirect dimension. Many of the residues involved were not observed in other dipolar-based spectra. A Lorentz-to-Gauss transformation was applied in both dimensions.

**Figure 8 viruses-14-01264-f008:**
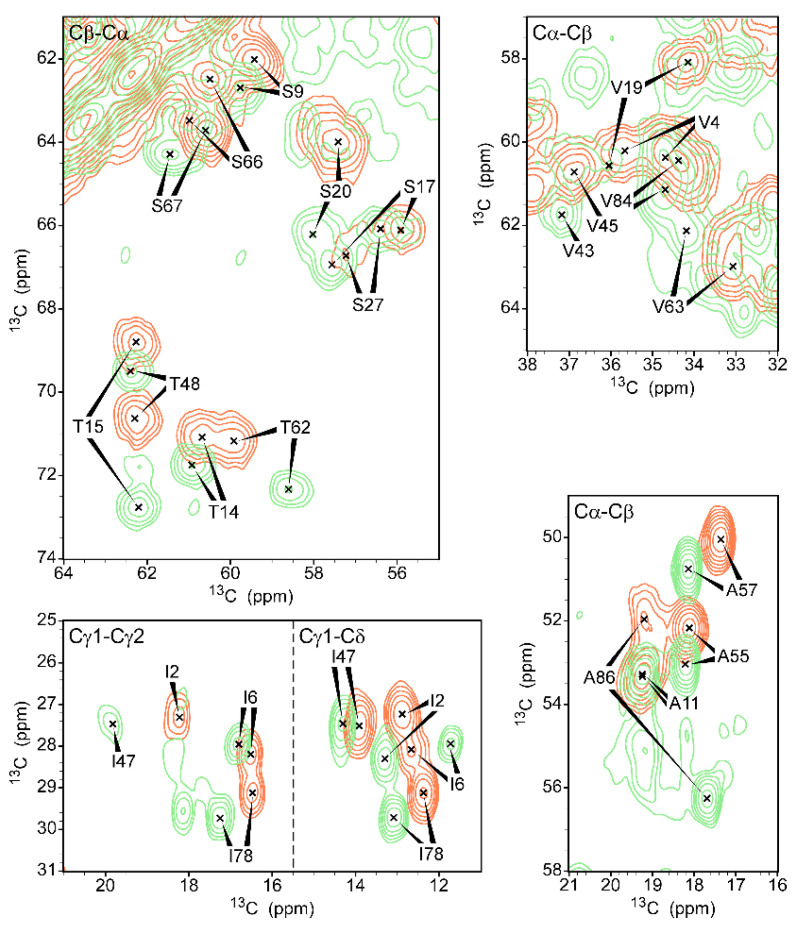
Comparison of several typical regions from ^13^C-^13^C DARR correlation spectra of free gVp (green, mixing time 100 ms) and ssDNA-bound gVp (orange, mixing time 15 ms). Fourteen contours were drawn at multiples of 1.4 with the lowest contour at a signal-to-noise value of 10. Clear spectral changes occur upon complexation across the entire protein.

**Figure 9 viruses-14-01264-f009:**
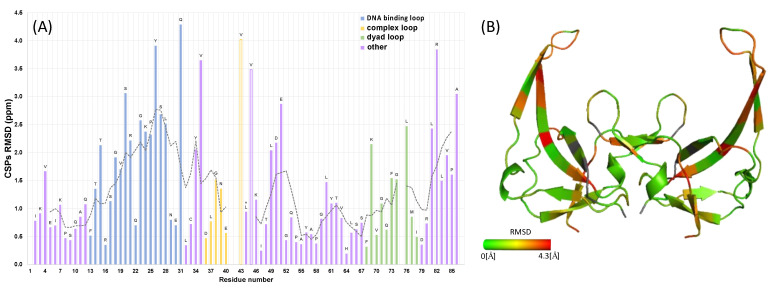
Weighted RMSD values extracted from chemical shift perturbations of free and bound gVp. (**A**) RMSD plot calculated by giving the ^15^N shifts one-quarter the weight of carbons. The different regions of the protein are colored as follows: blue, DNA binding loop (residues 13–31); yellow, complex loop (residues 36–43); green, dyad loop (residues 68–78); purple, remainder of the protein. A five-point moving average was plotted as a dashed line. There were no data for residues 1, 41, 42, and 87. (**B**) RMSD values plotted on the structure of the dimer structure from X-ray crystallography (pdb id 1vqb).

**Figure 10 viruses-14-01264-f010:**
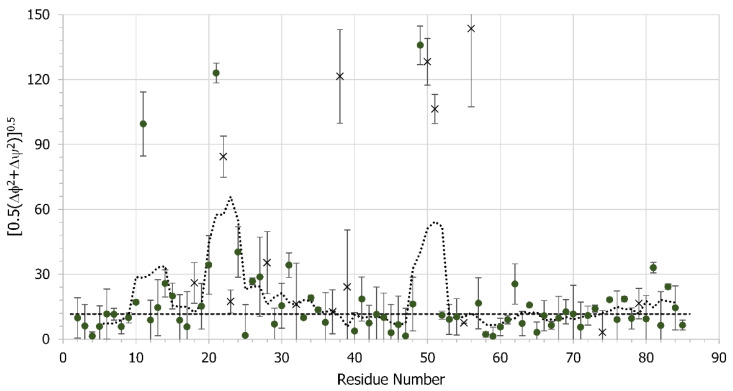
Quadratic average of the backbone torsion-angle differences $\Delta \phi =\phi_{free}-\phi_{bound}$ and $\Delta \psi =\psi_{free}-\psi_{bound}$, given by $\sqrt{0.5\left(\Delta \phi^{2}+\Delta \psi^{2}\right)}$, between the free and ssDNA-bound gVp. Predictions with low reliability in TALOS are marked with ‘x’. The dashed line represents the average of all entries having values < 30 degrees and equals 11.6 degrees. The dotted line is the 5-point moving average.

## Data Availability

Chemical shifts of gVp in complex with ssDNA can be downloaded from the BMRB (https://bmrb.io/, accessed on 1 June 2022) accession number 51391.

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
