# Peer review of "Conformational Changes in Ff Phage Protein gVp upon Complexation with Its Viral Single-Stranded DNA Revealed Using Magic-Angle Spinning Solid-State NMR"

_viruses, 2022, doi:10.3390/v14061264_

Round 1
Reviewer 1 Report
Although many methods have used to analyze the interaction between gVp and fp-ssDNA and some information was obtained, an accurate experimental atomic-resolution structure of the complex has yet to be established to investigate if and how the protein undergoes structural changes in order to facilitate DNA binding. In this study solid-state NMR was utilized to provide an insight into the structural changes that gVp undergoes upon the creation of the in vitro fd-gVp-ssDNA complex (consisting of a full-length ssDNA originating from the fd phage) by performing de-novo chemical shift assignment of the protein in the complex. Consequently, the authors compared these shifts to those of the free form, and compare the chemical-shift-derived secondary structure to the X-ray structure of a crystalline free gVp. This study is interesting, the experiments are designed well and the writing is well. Only some minor revisions are below:
1. Line31 phage is phagesï¼›
2. Line49 1500 is 1,500ï¼›
3. Figure 1 is from the reported study. Please be careful to use the images;
4. The image of Figure 2 showed some bright light especially in marker lane. Please avoid this;
5. I am not sure that the results and discuss are allowed to be together for this journal. Please confirm it.
6. In the conclusion part, the problem is repeated too much. Please keep concise.
Author Response
- Line31 phage is phagesï¼›
Corrected.
- Line49 1500 is 1,500ï¼›
We leave this to the editorial office. We haven’t since this style in other manuscripts of viruses.
- Figure 1 is from the reported study. Please be careful to use the images;
Figure 1 was created by us using pymol with coordinates available from the pdb. We now clarified this in the figure cation mentioning pymol. The pdb code is mentioned and the relevant paper is cited in the text.
- The image of Figure 2 showed some bright light especially in marker lane. Please avoid this;
We reduced exposure, and increased sharpness and contrast in order to make the gel image clearer. We also cropped the empty bottom part. We also added the ladder explicit values in the caption following the request of reviewer #3.
- I am not sure that the results and discuss are allowed to be together for this journal. Please confirm it.
The manuscript is significantly clearer in our opinion when the discussion is combined with the results, and we prefer to keep it this way, unless the editorial requests a change.
- In the conclusion part, the problem is repeated too much. Please keep concise.
The first section of the conclusion part has been revised to avoid unnecessary repeats.
Reviewer 2 Report
The manuscript titled “Conformational changes in Ff phage protein gVp upon com-2 plexation with its viral ssDNA: evidence from Magic-Angle-Spinning solid-state NMR” reported structural features of the full-length gVp-14 ssDNA particle. The results demonstrated how the gVp binds the ssDNA. For general recommendation, I found the paper to be overall well written and much of it to be well described.
Minors:
I would like to see a figure showing a structural model of U-gVp-NA-DNA complex presented in sticks or cartoon.
Page 1, Line 10: “ Escherichia Coli …” to “ Escherichia coli …”
Author Response
I would like to see a figure showing a structural model of U-gVp-NA-DNA complex presented in sticks or cartoon.
The reviewer probably refers to a potential model of gVp in complex with ssDNA. The labeling symbols (U-uniformly labeled, NA-natural abundance) have no effect on the structure. Our data is too preliminary to generate a clear model that is different or better from previous calculated models, which have been proposed before and cited in our manuscript. Therefore, we only point to the fact that the protein must undergo structural changes. We have recently acquired a significant amount of data that allows us to calculate a possible atomic-resolution structural model. This is out of the scope of the current manuscript, which proposes that significant structural changes are to be expected, and proposes their locations, but not how they are manifested as a 3D structure. The latter requires distance restraints and structure calculation (in progress in our lab). We therefore prefer at this stage not to propose another model.
Page 1, Line 10: “ Escherichia Coli …” to “ Escherichia coli …”
Corrected
Reviewer 3 Report
In the paper "Conformational changes in Ff phage protein gVp upon complexation with its viral ssDNA: evidence from Magic-Angle-Spinning solid-state NMR", Kedem et al. investigated the conformation change in fVp of the Ff phage with viral ssDNA by using special NMR techniques. The paper is well written, and the findings provided novel information of signifcant viral proteins. It could be accepted after minor revision:
(1) In the experimental section, the method of bacterial expression of the fVp protein and purification should be detailed.
(2) In Figure 2, the molecular weights of the Marker ladders should be indicated in the figure or in the figure legend.
(3) In Figure 3, imporant signals should be indicated by squares or arrows, and be introduced in the figure legend.
(4) The references are somewhat outdated. Some recent references should be cited.
(5) The advantage of Magic-Angle-Spinning solid-state NMR during exploration of viral protein structures should be emphasized and discussed as compared to other non-NMR methods.
Author Response
(1) In the experimental section, the method of bacterial expression of the fVp protein and purification should be detailed.
While this has been described in extensive details in a former publication from our group (see Biomolecular NMR assignment (doi: 0.1007/s12104-022-10076-5), and cited in the current manuscript, following the reviewer’s request we added to the Experimental section of the revised form the main expression and purification procedures.
(2) In Figure 2, the molecular weights of the Marker ladders should be indicated in the figure or in the figure legend.
We added the dsDNA ladder explicit values in the caption. Following a request from reviewer #1, we also slightly imaged processed the figure – we reduced exposure, and increased sharpness and contrast in order to make the gel image clearer. The original image is with the editorial office and is available on request.
(3) In Figure 3, imporant signals should be indicated by squares or arrows, and be introduced in the figure legend.
We marked the identity of the main signals of the ssDNA that are visible in the spectrum – ribose carbons, thymine resonances and some of their correlations. This is now indicated in the caption. Important protein signals are discussed extensively in the text and are marked in the Figures in section 3.2 and 3.3.
(4) The references are somewhat outdated. Some recent references should be cited.
The most relevant references refer to structure features and binding properties of gVp. These studies have been carried out up to the 90’s. Efforts have stopped since adequate methods were of too low resolution. Therefore, most references are 20-30 years old. We did find one interesting study from 2015 showing how Ectoine affects the secondary structure of gVp and reduces its binding affinity to ssDNA (10.1021/acs.jpcb.5b09506). This is now cited in the revised manuscript. Two additional manuscript we discuss in the revised version deal with the ability of gVp tethered to gold surface to bind non-complementary ssDNA strands, and how this binding is abolished in the presence of a complementary ssDNA strands that has a stronger affinity to the original ssDNA sequence (citations: 10.1016/j.cplett.2012.03.017 - 2012, 10.1021/la803596q - 2009).
(5) The advantage of Magic-Angle-Spinning solid-state NMR during exploration of viral protein structures should be emphasized and discussed as compared to other non-NMR methods.
We added a short paragraph in the introduction regarding advantages of ssNMR: “Magic-angle spinning solid-state NMR spectroscopy provides detailed atomic-resolution information on proteins, polynucleic acids, and protein complexes [38–40]. Magic-angle spinning solid-state NMR is particularly advantageous when studying systems of high molecular weight, or of low solubility, or systems that do not result in high quality crystals or that cannot be clearly resolved in cryo-electron microscopy. Solid-state NMR is especially useful for system that exhibits dynamics or intrinsic disorder that prevent characterization by diffraction techniques. These properties make solid-state NMR an excellent technique for studying intact viruses, viral capsids, and nucleocapsids [41–44].”